# Effect of Ultrasound Application on Protein Yield and Fate of Alkaloids during Lupin Alkaline Extraction Process

**DOI:** 10.3390/biom10020292

**Published:** 2020-02-13

**Authors:** Luis Alberto Aguilar-Acosta, Sergio O. Serna-Saldivar, José Rodríguez-Rodríguez, Anayansi Escalante-Aburto, Cristina Chuck-Hernández

**Affiliations:** 1School of Engineering and Sciences, Tecnológico de Monterrey, Monterrey N.L. 64849, Mexico; luis.raliugacosta@gmail.com (L.A.A.-A.); sserna@tec.mx (S.O.S.-S.); jrr@tec.mx (J.R.-R.); 2Department of Nutrition, Universidad de Monterrey, San Pedro Garza García N.L. 66238, Mexico

**Keywords:** *Lupinus* spp., protein isolates, alkaloids, protein extraction, ultrasound

## Abstract

The objective of this work is to elucidate the fate of quinolizidine alkaloids (QA) during the lupin protein extraction process assisted with ultrasound and the evaluation of the nutritional and functional properties of the protein fraction. Proximal characterization, concentration of anti-nutritional compounds, amino acid profile and protein solubility profile of flours from three lupin species were (*L. albus*, *L. angustifolius* and *L. mutabilis*) assessed. The result showed a significant difference (*p* < 0.05) in protein concentration, fat, total alkaloids and particle size between the three species flours. Based on these parameters, the most different *Lupinus* species (*L. mutabilis* and *L. angustifolius*) were chosen to study the behavior of the protein fraction in terms of functionality, composition and resistance to thermal treatments. The results obtained for *L. mutabilis* described the ultrasound effect as beneficial for protein yield (14% more than control), QA reduction from bagasse (81% less than control) and protein isolate production (50% less than control). On the other hand, *L. angustifolius* was more resistant to the ultrasound effect with no significant difference between treatments (10 and 15 min) and control but with the lower toxicity and better amino acid score. These results will be useful to design processes to assist in the objective of meeting the future protein demand of the population.

## 1. Introduction

Legumes are one of the most promising alternative sources of protein used for human and animal nutrition. As the world population grows, the demand for food with excellent aesthetic and organoleptic appeal derived from non-animal products increased e.g., simulated meats. Ingredient proteins should have acceptable intrinsic properties and the essential functional features for the variety of intended applications [1]. The functional properties of proteins that are relevant to food production are related to their physicochemical, structural properties and adaptability of domain structures of the whole molecule to changes in environmental conditions [2]. Environmental conditions are divided into interactions with food components (water, ions, proteins, lipids, carbohydrates, flavors) and the immediate environment, i.e., temperature, pH, and ionic strength. These features are directly affected by different factors during extraction, further uses and processing. The maintenance or development of the functional properties rely on proper isolation method and additional storage conditions [3]. Vegetable protein isolates are defined as food ingredients manufactured by solubilization and extraction of protein from ground seed particles. Extraction is a complex process composed by the following phenomena: (1) entry of the solvent into the particle; (2) redistribution of solvent in cell compartments and expansion of the solid matrix; (3) solubilization and/or degradation of components; (4) transport of the solute to the exterior of the particle and (5) migration of the extracted solute from the surface of the particle into the bulk [4]. The rate of protein extraction is usually controlled by processes occurring inside the particle, rather than by external factors, which makes microstructural modifications important [5]. In practice, size reduction is used to obtain high extraction rates; but this results in high expenditure of energy during milling, and difficulty in downstream separation of the protein solution from a fine residue. Therefore, a basic understanding of the effect of particle size and microstructure is desirable for designing efficient processes [4]. Since 1970, one of the most globally produced legume is soybean; having the highest amount of protein among legumes [6]. The seed with a more similar composition to soy, mainly in terms of protein, is the one produced by the plant of the genus *Lupinus* spp. better known as bluebonnet [7]. Some of the lupin varieties are toxic, due to their quinolizidine alkaloid (QA) content. These compounds are secondary metabolites, since their concentration is dependent on the species, cultivar, time and place of cultivation. Three species of the *Lupinus* genus native to the Mediterranean region; selected and domesticated as sweet lupins, are grown around the world: white or albus (*L. albus*), narrow-leafed (*L. angustifolius*), and yellow lupin (*L. luteus*). The pearl lupin (*L. mutabilis*) is native to South America, where it is grown, but considered as a bitter species due to high level of alkaloids. The sweet lupins (*albus*, *angustifolius*, and *luteus*) were selected by breeding because of its low alkaloid content (less than 200 ppm) and it is used as a valuable foodstuff requiring minimum prior treatment [8]. Production of protein isolates from lupin can be as a result of the high protein concentration of the legume. In order to overcome the problem of toxicity, the solubility properties of alkaloids can be used to remove them, and a polar fraction rich in these metabolites can be obtained and used as functional ingredients in human foods [9]. About seventy (70) different QA are present in the *Lupinus* species [10], most of which are in form of a salt, but twenty-eight (28) are free bases [11]. They have lethal hepatotoxicity and neurological effect on children in concentrations ranging from 10 to 25 mg/kg of body weight per day. The bitter alkaloids as named, ranges from 0.07 to 4.50 mg/g (dry weight, DW) in the lupin seeds [12], meaning that a child weighing 20 kg just need to eat about 4.44 to 285.71 g of raw lupin to reach the minimum lethal dose of total alkaloids. Despite the high toxicity, the bitter lupin seed has been consumed in the past 1500 years in the Andean region of South America after a debittering process with water (around 64 times water the dry weight of the seed) followed by a day cooking, soaking, and washing. The traditional method to remove the toxic alkaloids also resulted in the loss of up to 22% of the dry weight of the seed.

Many efforts have been made to solve the problem of lupin toxicity due to QA. The known debittering treatments belongs to one of the three main processes: (1) aqueous extraction; (2) biological degradation and (3) chemical extraction. In search of new technologies that do not cause the mentioned effects on nutritional and functional properties, the development of non-thermal processing methods to modify food ingredients has generated interest in the food industry recently. In particular, the use of high-intensity ultrasound (US) has attracted considerable attention due to its potential in the development of novel, gentle but targeted processes to improve the quality and safety of processed foods. US is an acoustic wave with a frequency greater than 20 kHz, the threshold for human auditory detection [13]. Based on frequency range, US can be classified into two distinct categories: (1) high frequency (100 kHz to 1 MHz) low power (<1 W cm^−2^) US and (2) low frequency (20–100 kHz) high power (100–1000 W cm^−2^) US. High frequency-low power US is commonly used for the analytical evaluation of the physicochemical properties of food, whereas low frequency-high power US has been recently employed for the alteration of foods, either physically or chemically [14]. Furthermore, the technology offers the potential of improving existing processes when used to support traditional techniques [15]. It is interesting to note that although the alkali-based extraction step in vegetable protein isolates production can be useful as a chemical alkaloid extraction method, there is no report using both steps in a procedure for bitter lupin species. In addition, the use of US as assistance in the production of lupin protein isolate has not being tested, even though it can enhance the physical separation of alkaloids from the protein storage bodies within lupin cotyledon. Thus, the aim of this research is to evaluate the fate of alkaloids and the protein yield from lupin, when used as a raw material for protein isolate production, and testing an alkali-based protein extraction procedure with the assistance of US as a novel technique to enhance production.

## 2. Materials and Methods

### 2.1. Raw Material

The lupin seeds (Figure 1) were obtained from the Centro Universitario de Ciencias Biológicas y Agropecuarias of the Universidad de Guadalajara (Guadalajara, Jalisco, Mexico).

### 2.2. Seeds Quality

According to the Government of Alberta, one thousand (1000) seed weight was measured using one thousand (1000) kernel weight for calculating seeding rates and harvest losses [16]. Test weight was measured using AACC 55-10 Test Weight per Bushel method [17]. Foreign material, damaged kernel and foreign seeds were measured by a gravimetric method using 100 g of lupin’s seed [18].

### 2.3. Scanning Electron Microscopy

The lupin seeds were frozen with liquid nitrogen and then cracked by half. The frozen seeds were mounted on a carbon double tape and coated with a thin gold layer (5 nm) before observing under a Scanning Electron Microscope (SEM) (Vega3, TESCAN, Czech Republic) equipped with a secondary electron detector at 1500× resolution.

### 2.4. Flour Characterization

Lupins’ flour was obtained using two mills: a knife mill (Wiley Mill, Arthur Thomas, Dover, NH, USA) and a sample mill (Cyclone Sample Mill, UDY Corporation, Fort Collins, CO, USA). Oil extraction was made using lupin’s flour and n-Hexane as a solvent in a ratio of 3:1. The extraction process was carried out at 50 °C in an orbital shaker incubator (RF 1575, VWR, Radnor, PA, USA) at 120 rpm for 48 h with one 24 h solvent change. After oil extraction, the desolvation of the flour was carried out at room temperature for 24 h. Then, a mild milling process in a coffee and spice grinder (GX410011, Krups, Solingen, Germany) was performed to homogenize the flour.

#### 2.4.1. Particle Size

Particle size was analyzed using a laser particle size analyzer (Mastersizer 2000E, Malvern Panalytical, Malvern, UK) coupled with a dry powder feeder (Scirocco 2000M, Malvern Panalytical, Malvern, UK).

#### 2.4.2. Proximate Analysis

The proximate analysis on the lupin’s flour was carried out by triplicate with the following methods. The crude fiber was determinate by the AOAC official method 962.09 [19]. Protein analysis was performed by following the micro Kjeldahl procedure under the AOAC official method 978.02 [19] using the conversion factor 6.25. Crude fat determination on dry lupin flour was made by the Goldfish protocol, using the AACC official method 30-20.01 [20]. Ash determination was made using a gravimetric method following the AOAC official method 923.03 [19]. The moisture analysis was carried out using the gravimetric method reported by the AOAC official method 925.10 [19]. Total carbohydrates were obtained by difference.

#### 2.4.3. Osborne’s Solubility Profile

The solubility profile of lupin’s flour protein was carried out by following the methodology proposed by Serna-Saldívar [18]. Protein determination of each fraction was made following the micro Kjeldahl method described in Section 2.4.2.

#### 2.4.4. Amino Acid Profile

The amino acid profile of the lupin’s flour and protein isolates were characterized by an external laboratory (Experimental Station Chemical Laboratories, Missouri University) according to the AOAC official method 982.30 [19].

#### 2.4.5. Electrophoretic Profile

Electrophoretic characterization was performed by SDS-PAGE under reducing and non-reducing conditions using a procedure described by Sambrook & Russell [21] with slight modifications. Protein bands were stained with Coomassie Brilliant Blue R-250 (Bio-Rad, Hercules, CA, USA), and the analysis of the gel images was performed using the GelAnalyzer 2010 software.

#### 2.4.6. Quinolizidine Alkaloids Concentration

The concentration of QA in *Lupinus* flour was determined by a titration method proposed by Ruiz et al. [22], using one gram (1 g) of defatted flour as sample. 

#### 2.4.7. Secondary Structure Analysis by FTIR

Fourier Transform Infrared Spectroscopy (FTIR) was used to elucidate the protein secondary structure of the lupin’s flour and isolates. Data were collected using an equipment ATR-FTIR (Perkin Elmer, Spectrum 1, Norwalk, CA, USA), following the procedure described by [23] and analyzing Amide I region (1600–1690 cm^−1^), correspondent to the protein secondary structure, according to Jackson & Mantsch [24].

### 2.5. Effect of Ultrasound in Lupins Protein Extraction and the Properties of Isolates Produced under Alkaline Conditions

The protein extraction was performed by the methodology proposed by Salinas-Valdés et al. [25], which consists of an alkaline extraction followed by acid precipitation. The complete methodology is presented in Figure 2. US treatments were performed in a 1000 mL beaker with approximately 550 mL of flour dispersion (1:10, ratio flour:water) in an equipment UP400S (Hielscher Inc., Teltow, Germany) with a 22 mm diameter cylindrical sonotrode (Horn 22, Hielscher Inc.) immersed 4 cm in the slurry. Samples were processed at a frequency of 24 kHz, 100% amplitude (100 μm), acoustic power density of 85 W/cm^2^ and pH 9.0, as described in Table 1.

#### 2.5.1. Protein Yield of Alkaline Extraction Process Assisted by Ultrasound

The three fractions of the extraction process (bagasse, protein curd, and serum) were obtained, weighted and all were analyzed for moisture and protein percentage following the procedures described in Section 2.4.2. Protein yield was then calculated employing the Equation (1).
(1)$$Protein\;Yield=\frac{Fraction\;Protein\;Weight\;\left(g\right)}{Flour\;Protein\;Weight\;\left(g\right)}\;\times \;100$$

#### 2.5.2. Functional Properties of Protein Isolates Obtained from Ultrasound Assisted Alkaline Extraction Process

The functional properties evaluated were Water Absorption Index (WAI) [26], Water Solubility Index (WSI), Nitrogen Solubility Index (NSI) [27], and in vitro protein digestibility [28]. Purity was calculated as the ratio between protein in the protein isolate and the total weight of the isolate expressed in 100 bases. PD-CAAS (Protein Digestibility Corrected Amino Acid Score) was represented as the percentage of the limiting essential amino acid covered by protein isolates multiplied by the in vitro protein digestibility. 

#### 2.5.3. Amino Acid Profile, Electrophoretic Profile, and Secondary Structure of Protein Isolates Obtained from the Ultrasound Assisted Alkaline Extraction Process

These evaluations were assessed according to the methods described in Section 2.4.4, Section 2.4.5, and Section 2.4.7, respectively.

#### 2.5.4. Quinolizidine Alkaloids Concentration in Serum, Bagasse, and Protein Isolates Obtained from Ultrasound Assisted Alkaline Extraction Process

QA concentration of protein isolates from *L. mutabilis* was determined by the same protocol used for the *Lupinus* flour in Section 2.4.6. QA concentration of protein isolates from *L. angustifolius* was obtained with Capillary Gas Chromatography (HP6890 Plus GC, Agilent Technologies Inc., Santa Clara, CA, USA) coupled with a mass selective detector (5973N, Agilent Technologies Inc., Santa Clara, CA, USA) (GC-MS). This determination was carried out with a 0.25 mm × 30 m × 0.25 μm (5%-Phenyl-methylpolysiloxane capillary column 5-HP, Agilent Technologies Inc., Santa Clara, CA, USA). The mobile phase at 1 mL/min flow rate was helium. The alkaloid extraction for GC-MS was carried out by following the method proposed by Muzquiz et al. [29], about 0.5 g of defatted material with a solution of known concentration of strychnine (S0532, Sigma Aldrich, St. Louis, MO, USA) on dichloromethane was used. In the case of protein curd and bagasse, they were freeze-dried for 48 h and milled. After evaporation, the sample was suspended with fresh dichloromethane and taken to a volumetric flask. The GC-MS applied an injection volume of one (1) μL.

### 2.6. Statistical Analysis

The statistical analysis was performed using the software Minitab^®^ 18. An analysis of variance –ANOVA- was carried out to evaluate the effects of the experimental factors, with 95% of confidence. The Tukey analysis was applied at *p* < 0.05 to assess differences using the specific treatments.

## 3. Results

### 3.1. Seed Quality

The parameters of seed quality evaluated from the three varieties of *Lupinus* are presented in Table 2. According to the Australia Pulse Standard [10], the three species are commercial grade. These results prove the similarity of the raw material used in this study with the market product.

### 3.2. Scanning Electron Microscopy of Lupinus Seeds

In order to understand the *Lupinus* seeds’ physical features, the microscopic analysis was performed to observe the structural characteristics of the seed coats as shown in Figure 3. 

There were visual differences among the species; the morphology of the osteosclereids layer of *L. mutabilis* was more relevant. The *L. angustifolius* seed had the thickest coat, and this could be related to the low alkaloid content since primary functions of QA include nitrogen transport and chemical defense against herbivores. Therefore, the strength of the seed as a protection mechanism against pest attack due to low alkaloid content could be the reason why the sweet lupin had the thickest pericarp [30].

### 3.3. Characterization of Flours

#### 3.3.1. Particle Size

The particle size median (P_50_) of lupin flour for the three species; *L. albus*, *L. angustifolius*, and *L. mutabilis* were 48.93 ± 0.31; 178.42 ± 0.47, and 44.36 ± 3.26 μm, respectively. Since the milling procedure for all samples are the same, differences in particle size between species depicted variations in the strength of the seed, as well as differences in moisture and oil concentration. As a result of this, protein extraction differences among lupin varieties are foreseen.

#### 3.3.2. Proximate Analysis

Significant differences for the three species were detected in the proximate analysis (Figure 4). Compared with Carvajal-Larenas [12], the overall results coincide with each species profile. The variety with highest protein and fat concentration was *L. mutabilis*, with a similar composition to soybean; this is the variety previously reported with the highest commercial and nutritional potential.

#### 3.3.3. Profile of Osborne Solubility

As shown in Figure 5, 58%, 85% and 60% of total protein for *L. mutabilis*, *L. albus*, and *L. angustifolius*, respectively are associated with two Osborne’s fractions: albumins and globulins, data that gives an insight about the extractability of the protein using traditional isolation techniques. 

#### 3.3.4. Amino Acid Profile

The amino acid profile is vital for the assessment of the nutritional potential of vegetal proteins. The results of the three varieties are in Table 3. 

The essential amino acid with the lowest concentration was methionine, there was no significant difference between the species. These results were compared with the Carvajal-Larenas [12], there were a few different values, for example, the concentration of valine reported for *L. albus*, *L. angustifolius,* and *L. mutabilis* were 3.8, 3.7 and 3.8 g/100 g protein, respectively. Meanwhile, the results obtained herein were 4.01%, 4.48% and 4.05% for the three (3) varieties. Protein percentage and amino acid profile (especially methionine), are the two compositional characteristics mostly affected by cultivar, cultivation year and season as well as the soil and nutrients [31].

#### 3.3.5. Electrophoretic Profile 

In Figure 6, the SDS-PAGE of raw *Lupinus* flours was presented. The profile depicts differences among samples, *L. mutabilis* being the cultivar with the highest number of bands in the upper part of the gel (high molecular weight), while *L. albus* seems to be the variety with a slight deviation down the ladder. In non-reducing conditions, a 20 kDa band in *L. angustifolius* appears to make the difference among all samples. This particular band is preceded by a 25 kDa in all three cultivars followed by several between 10 and 20 kDa.

#### 3.3.6. Quinolizidine Alkaloids Concentration

QA are the most toxic anti-nutritional factors of *Lupinus* seeds. The *Lupinus* flours QA concentrations obtained by titration of lupanine were 0.16, 0.04, and 1.75 g/ 100 g. DW in *L. albus, L. angustifolius* and *L. mutabilis*, respectively. The latter variety statistically showed the highest QA concentration. 

#### 3.3.7. Trypsin Inhibitor Activity

The analyzed lupin flours did not have detectable trypsin inhibitor activity (TIA). Nevertheless, the protein extraction process may perhaps concentrate this factor due to their protein nature. Meanwhile, these results offer an advantage over other legumes like soybean which has about 30.1 Trypsin Units Inhibited (TUI)/mL [32].

#### 3.3.8. Secondary Structure by FTIR

The secondary structure of proteins from *Lupinus* flours are summarized in Table 4. They are presented in the percentage of total Amide I peak. The native protein of legumes has a secondary structure dominated by β-sheets and β-turn structures. Secondary structures are responsible for functional, nutritional and nutraceutical properties of protein and there is evidence that revealed the effect of thermal processing over the secondary structure [33].

### 3.4. Ultrasound Effects in Lupin Flours Protein Extraction Process, and Properties of the Obtained Isolates

#### 3.4.1. Protein Yield of US Alkaline Extraction Using *L. mutabilis* and *L. angustifolius*

In Figure 7, are presented the protein yield of each fraction (protein curd, bagasse, and serum) derived from the three protein extraction processes (control, US-assisted 10 min, and US-assisted 15 min) over *L. mutabilis* and *L. angustifolius*. 

For *L. mutabilis* samples, significant statistical differences were observed between protein curd and bagasse. The results of *L. angustifolius* showed that there was no significant difference between treatments on protein yield of the fractions. Nevertheless, the highest mean protein yield for the protein curd was achieved with the most severe treatment (15 min).

#### 3.4.2. Functional Properties of Protein Isolates from *L. mutabilis*, and *L. angustifolius* Obtained from US-Assisted Alkaline Extraction Process

In Figure 8 are summarized the functional properties of protein isolates from two *Lupinus* varieties.

WAI of *L. mutabilis* protein isolate obtained without US (control) was about 1.5 g water/g isolate similar to results reported by Carvajal-Larenas [12]. The WSI results of protein isolates compared with the control were lower; both treatments (10 and 15 min) showed half the solubility of control. The Nitrogen Solubility Index (NSI) for control was about 50%, which is higher compared with US treatments (10 and 15 min), where a loss of half NSI was observed. In vitro protein digestibility was also assessed, and the result showed that all samples (control, 10, and 15 min treatments) reached almost a 100% digestibility with no statistical differences. The percentage of purity of the samples from *L. mutabilis* isolates showed statistical differences between the control and the 15 min US treatment, with lower purity in the 15 min US treatment. The reported PD-CAAS from *L. mutabilis* is around 0.7 [12,34]. This data corresponds with the obtained from the US-assisted alkaline extraction: 0.67 for the control and 10 min treatment isolates and 0.65 for the 15 min treatment. For *L. angustifolius*, despite there was no significant difference among the ultrasound treatments and the control regarding the WAI, the 10 min US treatment had the highest mean. Compared to *L. mutabilis*, the *L. angustifolius* protein isolates had lower water absorption. WSI of *L. angustifolius* was lower than *L. mutabilis* protein isolates because of the high insoluble fiber present in the raw seed. There were no significant differences between US treatments and control. The same reduction when compared with *L. mutabilis* isolates was observed for NSI. When examining the in vitro protein digestibility of the isolates from *L. mutabilis* with *L. angustifolius*, both were up to 90% with no substantial differences between them. The percentage of purity in *L. angustifolius* samples showed no effects of the US treatment, but lower values were found when compared with isolates from *L. mutabilis*. The PD-CAAS for control, 10, and 15 min US treatments were similar among them, but with a slight tendency to increase with the US time (15 min US treatment had the higher PD-CAAS: 0.82).

#### 3.4.3. Electrophoretic Profile of *L. mutabilis* and *L. angustifolius* Isolates Obtained from US-Assisted Alkaline Extraction Process

The electrophoresis pattern of the protein isolate obtained from *L. mutabilis* and *L. angustifolius* are illustrated in Figure 9. Protein structure derived from the control treatment of *L. mutabilis* flour did not have significant changes compared with the native seed protein (Line 2 and 1 from Figure 6, respectively). Nevertheless, the 10 and 15 min showed a group of higher molecular weight bands between 250 and 50 kDa. Also, two bands appeared; one of 37 and other from 31 kDa.

*L. angustifolius* protein isolates showed no significant difference between treatments and control except for the intensity in certain bands that could be due to different protein concentrations (Figure 9b). This result explains the similarity in behavior of the protein isolates from the different treatments.

#### 3.4.4. Amino Acid Profile of *L. mutabilis* and *L. angustifolius* Protein Isolates Obtained from US-Assisted Alkaline Extraction Process

The main interest in *Lupinus* varieties of food is related to its high content of protein, which is considered a good source of lysine (especially *L. mutabilis*) but generally deficient in the sulfur-containing amino acids [9]. Table 5 shows the amino acid profile and amino acid score for *L. mutabilis* protein isolates obtained for control, 10, and 15 min of US-assisted extraction. 

The main difference between the amino acid profile for flour and protein isolates of *L. angustifolius* (Table 5) was the enhanced nutritional properties, because of the concentration of five (5) essential amino acid due to the US treatment.

#### 3.4.5. Trypsin Inhibitor Activity of *L. mutabilis* and *L. angustifolius* Protein Isolates Obtained from US-Assisted Alkaline Extraction Process

The protein isolates did not have TIA in both varieties. 

#### 3.4.6. The fate of Alkaloids during *L. mutabilis* and *L. angustifolius* Ultrasound Process for Protein Extraction

The final concentration of alkaloids (fate) in every fraction obtained from the *L. mutabilis* and *L. angustifolius* protein extraction is presented in Figure 10. 

The blue fraction represents the percentage of alkaloids obtained in the protein curd for control, 10, and 15 min US-assisted treatments with 0.06%, 0.03%, and 0.04% of lupanine DW of alkaloids, respectively. The 10 min treatment had the lowest alkaloid content with 30 mg lupanine/100 g protein isolate, for example, 58 times less alkaloid concentration than the raw flour (1750 mg/100 g *L. mutabilis* flour) and half the concentration compared with the traditional (control) protein curd. The bagasse fractions for the control, 10, and 15 min US treatment were 0.49%, 0.09% and 0.08% of lupanine DW, respectively, also depicting a reduction in the toxicity compared to the control due to the US effect. The 15 min treatment had six times few alkaloids than the control. The 80 mg of lupanine/100 g of bagasse fraction is more than 21 times less toxic than *L. mutabilis* flour. The US treatment had adverse effects on this parameter when *L. angustifolius* samples were evaluated, increasing the alkaloid content on the protein isolates. Conversely, the QA concentration on bagasse and serum was lower in US treatments than the control. The alkaloid concentrations in the protein curd were 1.6, 4.2, and 6.1 mg of lupanine/100 g DW, this represents a decrease of 94, 85, and 78% for control, 10, and 15 min US treatments applied to *L. angustifolius* flour, respectively.

#### 3.4.7. Secondary Structure of Protein Isolates from *L. mutabilis,* and *L. angustifolius* Produced with Ultrasound-Assisted Extraction Procedure

The secondary structure of the *Lupinus* protein from the isolates produced with and without US-assisted extraction is presented (Table 6). The main difference among treatments and the native protein (Table 6) was the presence of aggregated strands that increased according to the US time.

## 4. Discussion

### 4.1. Seeds Quality and Morphology

The one thousand (1000) seed weight is an indicator of the size of the seed and is often used to calculate the crop densities to obtain better crop yield. This number is highly variable even between the same species [16] and due to factors such as harvest year and soil conditions. In the case of the three lupin’s cultivars, results were similar to those obtained for Gdala & Buraczewska [35]. The thickness and morphology of the cell wall are relevant in the strength and water absorption capacity of seed. This layer is responsible for two crucial processes during seed development and germination: water evaporation and water absorption. The former allows the seed to achieve the dormancy state due to dryness. The latter, contributes to the debittering processes based on aqueous extraction [31], as well as help in the distribution of water during germination. The thickness of the coat is relevant for the milling and cooking process. In addition, only 1%–7% of total alkaloids are present in the pericarp or hull of the seed, depending on the species.

### 4.2. Flour Characterization

The milling protocol was designed to obtain the minimum particle size since it correlates directly with the extraction efficiency. The value P_50_ is the highest diameter (in μm) of the 50% of the particles within the flour, and this value is crucial to determine the efficiency of the oil, protein, and alkaloid extraction in terms of time and yield [12]. The milling process depends on the moisture of the seed, oil concentration and strength of the seed coat. The SEM analysis showed that *L. angustifolius* seeds had the thickest pericarp (Figure 3b), which was related to the larger particle size of flour of this variety. In the case of proximal composition, the only difference worth to be enhanced is the *L. angustifolius* fiber content, higher than *mutabilis* variety (around 14% vs. 6%, respectively). 

According to Osborne solubility, from the five fractions, the non-extractable was in the last group in Figure 5 (residue). It is evident that *L. mutabilis*, despite its high percentage of protein, about 30% stayed at the end of a high and intensive protocol, so it will not be available for extraction and protein isolates production using conventional techniques. Varieties *albus* and *angustifolius* seem to be good alternatives for industrial extraction process despite seed’s physical differences and different protein conformation. 

Regarding the amino acid profile, the concentration of the sulfur-containing amino acids was low. The recommended dietary allowance for both cysteine and methionine are 58, 27, 22, and 13 mg/kg per day depending on the age group (infants < 6 months old, 2-year old children, 10–12 years old, and adults respectively). The results obtained for *L. albus*, *L. angustifolius*, and *L. mutabilis* were 2.50, 2.58 and 2.20 g/100 g protein, respectively. Taking that into consideration, the infant must consume about 50 g of *L. mutabilis* seed (protein: 40% dry weight) in a day to achieve the dietary requirements [36]. 

The SDS-PAGE analysis allows the study of the globulins, which are the main fraction of lupin’s seed storage proteins since they comprise the most nutritional, technological, nutraceutical and allergenic potential. Globulins are the salt-soluble storage proteins and can be divided into four (4) groups, depending on its electrophoretic mobility, α-, β-, γ- and δ-conglutins. 

There are few reports regarding the electrophoretic pattern of lupine, and this information is very useful, more than just for characterization, to create strategies for protein separation and fractioning. The size and weight of each fraction give an idea of the centrifugal force needed to separate the protein curd from the serum. Also, the isoelectric point is a little bit different among fractions and species and could cause the loss of specific portions crucial to nutritional and functional properties. 

Regarding QA concentration in *Lupinus* flour and considering a maximal QA daily dose for adults of 0.35 mg/kg body weight, a 75 kg grown man can consume up to 26.25 mg of QA per day [10]. The lupin’s flour needed to reach the daily dose is 16.4, 65.6 and 1.5 g of *L. albus*, *L. angustifolius* and *L*. *mutabilis*, respectively. According to these results, *L. albus* and *L. angustifolius* are considered sweet species due to the low concentration of QA compared to others like *L. mutabilis* which is considered a bitter one.

Lupin flours obtained from the three varieties showed no detectable TIA, this partially agrees with a study conducted by Schoeneberger et al. [32] where five ecotypes of *L. mutabilis* were analyzed, and four (4) did not show any TIA. Only one ecotype presented 1.16 TUI/mL; this was 30 times less than soybean. According to Erbas et al. [37], the overall lupins inhibitor activity is shallow, and the reported concentration is in the range of 0.1 to 0.2 mg/g. 

Fourier Transform Infrared (FTIR) Spectroscopy is an established tool for the characterization of protein secondary structure. Czubinski et al. [38] described the complete structure of a 7S basic globulin; this is a two subunits monomer present in *L. angustifolius* called γ-conglutin (45.4 kDa). The secondary structure of this protein had a rare conformation formed by a 3_10_ helix/α-helix complex. In the FTIR deconvolution, a non-common percentage of 3_10_ helix was found, possibly indicating the presence of high concentrations of γ-conglutin. 

### 4.3. Effects of the US Treatment in the Protein Isolates Yield and Functional Properties

*L. mutabilis* and *L. angustifolius* flours only were used as raw materials for the protein concentration (isolation) procedure to achieve more evidence of the effect of the US treatment that was obtained by preliminary assays. The work was then done with the so-called sweet and bitter lupins (low and high alkaloid percentage). 

For industrial and commercial proposes, protein yield is significantly essential. The yield of the process is influenced by three principal factors: flour particle size, temperature and protein composition. Protein yield for *L. mutabilis* isolates the 10 min US treatment result in the highest protein yield extraction on protein curd. Karki et al. [39] found a direct correlation between US time and protein concentration on isolates made from soybean flakes. This phenomenon was attributed to the cell rupture caused by the cavitation that releases sugars and protein from the seed tissue. Nevertheless, the longest time proved was 120 s, where a 78 ± 0.40% of protein yield was obtained. For lupin, 10 min was the maximal US time treatment to enhance the protein release. The 15 min treatment was counterproductive in *L. mutabilis* isolates (Figure 7a), dropping the protein yield. This could be due to extreme protein structural damage, which produces aggregates that do not solubilize with rise in pH and possibly end in the bagasse (centrifugation residue), where the 15 min US treatment bagasse fraction was represented. 

For *L. angustifolius* isolates, the highest protein yield was found in the protein curd treated with 15 min of US. This result showed a different tendency when compared with the *L. mutabilis* extraction, where the highest yield was reached with the 10 min US-assisted process. Comparing the protein yield results with Muranyi et al. [40], who combine several methods like isoelectric point precipitation, salt-induced extraction, alkaline extraction, dilutive precipitation, and their combination over lupin flour the results obtained with US are higher. The most comparable result presented by those authors was a 31.7 ± 0.8% yield using alkaline extraction and a combination of dilutive and isoelectric precipitation. In this research, using US alkaline extraction, more than 50% yield was obtained compared with conventional alkaline extraction in combination with isoelectric precipitation. The lower yield of *L. angustifolius* protein fraction (compared to *L. mutabilis*) may be explained by the differences in the flour particle size (Section 3.3.1), the composition and structure of storage proteins.

The functional properties and technological value of the proteins may be influenced negatively by industrial processing. This issue may lead to non-enzymatic browning and secondary structural changes due to heat [7]. Even with the same amount of protein, two protein isolates can have different water absorption; this could be because of the different globulin fractions between species and cultivars. The reported lupin flour water absorption (*L. albus* and *L. mutabilis*) were 2.3 and 1.2 g water/g flour DW, respectively. The reason for low WAI value of *L. mutabilis* is because of the fat content and probably absence of δ-conglutin, rich in polar amino acids [12]. The most important effect of the US during the alkaline extraction for *L. mutabilis* process is the 15 min treatment, resulting in more than four times higher WAI than the control, probably due to the denaturalization of the protein during the procedure, which causes the exposition of polar amino acids in the medium. The polar amino acids can form hydrogen bonds that enhance the interaction with polar solvents like water [9]. The water absorbed by flour reported for *L. angustifolius* was 2.4 water/g flour DW [12], this result did not correspond with that depicted in Figure 8b and could be linked to the high carbohydrate concentration in flour. The lower WAI on *L. angustifolius* protein isolates could be linked to composition differences, glycosylation pattern and protein concentration [41]. The difference in protein concentration for *mutabilis* and *angustifolius* isolates can indeed affect the water absorption, mainly because in the case of *L. angustifolius*, the protein concentration was lower (Figure 8), indicating the presence of other compounds that affect directly to the protein swelling process [1]. In other words, to have a higher water absorption, the protein needs to change its structure and make space for the water molecules, but when other compounds as fiber are in the system, this absorption process can be altered.

WSI is a parameter that can be used as an indicator for the degradation of molecular compounds and it depends on the availability of hydrophilic groups that bind water and enhance gel formation of macromolecules. An increase in protein concentration could lead to the diminution of WSI due to the ligation with other soluble molecules [42]. The protein concentrates had a low concentration of oligosaccharides, which plays an important role in the water solubility of the product [43]. Other proteins like albumins can increase the WSI due to their affinity to water. Nevertheless, legumes have low concentration of albumins, and protein isolates obtained by isoelectric precipitation do not contain albumin, and the amount of protein-polysaccharide complex is meager. The WSI results of protein isolates of *L. mutabilis* compared with the control treatment were low; both US treatments (10 and 15 min) showed half the values of those found in the control treatment. The US treatment could lead to the protein denaturalization and the possible formation of non-soluble structures capable of interacting with other macromolecules present in the protein isolate. Besides, the US treatment can affect the flour carbohydrates too i.e., gelatinizing the starch leading to gel formations and insolubilization of previously soluble fiber. In general, WSI values in *L. angustifolius* isolates were the lowest, and no significant differences were found among US treatments.

The NSI describes the ratio between soluble nitrogen and total nitrogen. This parameter is affected by pH, temperature, and ionic strength of the protein solubilization media. For *L. mutabilis* isolated, the US treatment with 10 and 15 min showed a reduction of 50% than the NSI in the control treatment due to the thermal treatment caused by US at a pH higher than the isoelectric point. The temperature in the beaker was 63 °C and 77 °C for the 10 min and 15 min treatment, respectively. The protein isolate was not neutralized after freeze-drying; this also can drop the solubility of the proteins as more time is stored in their isoelectric point [44]. The explanation of this could be the irreversible formation of protein conglomerates and stable conformation with low solubility where the protein is trapped. Isolates of *L. angustifolius* showed lower values of *L. mutabilis*; however, the samples obtained with 10 min of US application presented the highest values when compared with the other processes. Only one report was found regarding NSI for *L. angustifolius* samples, where NSI was about 13.1% for flour and 19.2 to 33.8% for the isolates obtained with different pH levels through alkaline extraction [44].

The results from *L. mutabilis* isolates *in vitro* digestibility was significantly higher than the 71.07% and 75.05% reported by Sathe, Desphanade, & Salunkhe [45] when *L. mutabilis* flour and protein concentrate were assessed, respectively. The same authors described the effect of a 30 min boiling step of flour and protein extraction of this variety, yielding an increase of 4.63 and 3.81% in this parameter, respectively, probably due to the protein denaturation, which facilitated the enzymatic degradation. In vitro protein digestibility for *L. angustifolius* was reported as 80% and 86.3% for flour and protein isolate at pH 12 respectively [46]. These results are similar to the ones depicted in Figure 8b, where the null effect of US was cleared and no significant difference were found for the treatments. Protein Digestibility Corrected Amino Acid Score (PD-CAAS) is the most useful tool to determine the ability of certain proteins to proportionate essential amino acids to the human diet. It reflects the essential amino acid content, true protein digestibility and the bioavailability of the amino acids on the food [47]. The values of PD-CAAS obtained in *L. mutabilis* isolates (Figure 8a), are very close to those reported but apparently the higher the US time, the lower the PD-CAAS; not because of a low digestibility but due to reduction in cysteine (0.03 g/100 g of protein). In addition, a slight decrease in cysteine could be seen on 10 min US treatment, but this is compensated with the higher digestibility, and that is why this treatment did not present significant difference in the PD-CAAS when compared with the control. For *L. angustifolius* isolates, there were no differences among treatments, but there was a slight tendency to increase as the US time arises. This result was associated with a 0.23 g/g crude protein increase in methionine concentration beside a small increase on *in vitro* protein digestibility.

### 4.4. Effects of the US Treatment in the Protein Isolates Electrophoretic Profile

Change in composition and secondary structure may be expected after the protein isolation process due to physical and chemical damage as well as differences in the solubility of fractions. Filtration, dilution and isoelectric precipitation are the main techniques used to separate protein from the aqueous solution in order of less severity to more. According to Muranyi et al. [40], the most efficient in terms of yield is the last. The differences in solubility of the protein fractions could lead to concentration of certain proteins like globulins and albumins, and the discard of other lipophilic fractions. This results in a protein isolate with different functional, nutritional and nutraceutical properties. For *L. mutabilis* isolates, the high molecular weight bands can be produced by new protein aggregations and coagulation attributed to physical damage due to US treatment. The low molecular weight new bands can be attributed to protein denaturalization because of the internal heat produced during US treatment. In this case, the temperature used in the treatment of 15 min US-assisted was 77 °C. Isolates from *L. angustifolius* did not show noticeable, in the electrophoretic profile, differences among treatments; however, the presence of a band between 37–50 kDa can be attributed to γ-conglutin; a nutraceutical protein which has several activities such as hypoglycemic, anti-cholesterolemic and ACE-inhibitory activity [38,48,49]. 

### 4.5. Effects of the US Treatment in the Protein Isolates Amino Acid Profile

The main interest in lupin for foods is related to its high content of protein, which is considered a good source of lysine (especially *L. mutabilis*) but generally poor in the sulfur-containing amino acids. For *L. mutabilis* isolates, the most remarkable increase (10% compared with control) was for valine, methionine, tyrosine and phenylalanine, whereas serine and cysteine were reduced. In the samples of isolates obtained from *L. angustifolius* flour, the major effect was the increase (10% difference compared with control) for threonine, valine, alanine, methionine, lysine, and tryptophan whereas serine, glutamic acid and cysteine were reduced with US treatment. Loss of polar amino acids and concentration of non-polar amino acids were expected when aqueous processing was executed and protein agglomeration leads the underexposure of hydrophobic amino acids and the medium protecting them from physical or chemical damage. As a result of the US treatment, the main difference between the amino acid profile for flour and protein isolates was the enhanced nutritional properties because of the concentration of five essential amino acid.

### 4.6. Effects of the US Treatment in the Protein Isolates Trypsin Inhibitor Activity

US treatments did not show negative effects on this parameter. *L. mutabilis* protein isolates did not present TIA detectable. The same trend was shown in the *L. angustifolius* samples even when 1% of polyvinylpyrrolidone (PVP) was added (to avoid the phenolic compounds interference). 

### 4.7. Effects of the US Treatment in the Fate of Quinolizidine Alkaloid Presents in the Protein Isolates

In *L. mutabilis* isolates, the 10 min US treatment reduced about 50% the concentration of the alkaloids from protein curd fraction when compared with non-US alkaloid extraction (control). This means that an adult of 70 kg could have ingested about 8 kg of protein isolate, which is equivalent to almost 7 kg of crude protein per day with no adverse effects. The cavitation produced by US over the cells is responsible for the liberation of alkaloids in the solution, and this phenomenon, along with the pH changes leads the migration of most alkaloid compounds to the aqueous phase (serum fraction, Figure 10a). Serum resulted in having the highest alkaloid concentration. Compared to control, treatments had up to 41% more of these compounds. QA concentration on bagasse and serum was lower in the US treatments for isolates obtained from *L. angustifolius*; this phenomenon could be due to enhanced water absorption of the protein isolates because of the carbohydrate concentration of the final protein curd (Figure 10b). The US might have liberated the alkaloids disrupting seed cells and more glycoproteins, but being the latter capable of absorbing alkaloid rich solvent, this reduces the QA concentration in the serum during protein precipitation. The results showed in this research are crucial for determination of toxicity. The safe dose of total QA is 35 mg/kg/day for adults. Regarding this, an adult weighing 70 kg could safely consume (daily) 153, 58 and 40 kg for control, 10 and 15 min protein isolates, respectively. This amount of protein overcome any protein’s nutritional requirement (amino acids and total protein) [10].

### 4.8. Effects of the US Treatment in the Protein Isolates Secondary Structure

The alkaline solubilization could lead to changes in secondary structure; therefore, the protein isolates control samples also had aggregated strands in the *L. mutabilis* samples. The US time enhanced this phenomenon, and the 15 min treatment had the highest aggregated strand percentage. Unordered structures just appeared on US treatment; this finding might be the explanation for the changes in functional and electrophoretic properties of protein isolates. The unordered structures are low soluble and easily degradable by enzymes. The disappearance of 3_10_ helix could lead to the formation of that unordered structures due to the weak intermolecular forces. The 3_10_ helix loss decreases the nutraceutical potential of the protein isolates due to the possible denaturalization of γ-conglutin and the consequent loss of hypoglycemic activity 

## 5. Conclusions

The three *Lupinus* varieties showed substantial differences mainly in protein content, secondary structure, solubilization profile, electrophoretic pattern, and composition (amino acid profile). Due to this, distinct protein extraction performance was obtained, especially when the process was assisted with US. *L. mutabilis* and *L. angustifolius* were selected to obtain more evidence of the effect of the US treatment in a wide range of alkaloid and protein content within these seeds. *L. mutabilis* showed the highest protein concentration in flour. The electrophoretic profile revealed the presence of a high percentage of water-soluble fraction, while amino acid profile confirmed low cysteine and methionine concentration in both flour varieties. The effect of US application over protein yield was beneficial for *L. mutabilis* due to an increase of 14% with 10 min treatment, whereas for *L. angustifolius* isolates, there was no significant difference with the use of this technology. A remarkable result was that *in vitro* digestibility was not affected by the US treatment, although SDS-PAGE showed an increase in protein agglomerates due to US application during the alkaline extraction procedure. This result was also confirmed with the secondary structure analysis, where a high proportion of aggregated strands and unordered structures appeared in isolates obtained at 10 and 15 min. *L. mutabilis* protein isolates indicated a QA concentration reduction for both bagasse and protein isolate with 10 min US-assisted extraction. The opposite trend was observed for *L. angustifolius* because it was found an increase in QA content for both bagasse and protein curd fractions, which was associated with the US treatment. US application for the extraction of protein isolates from *L. mutabilis* resulted to be a useful tool due to increase in protein yield and functional properties associated with changes in protein structure. The association between US application at different times and functional changes described herein could help create new protein profiles with desirable solubility and water absorption. Moreover, the environmental and economic implications that comes with the use of this technology for lupin isolates production process are highly relevant in the pursuit of new protein sources since less water usage, processing time, waste production, and energy expenditure are the main advantages of US application.

## Figures and Tables

**Figure 1 biomolecules-10-00292-f001:**
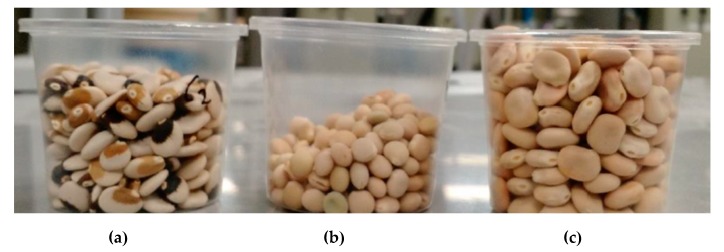
Lupin seeds from different varieties: (**a**) *Lupinus mutabilis*; (**b**) *Lupinus angustifolius*; (**c**) *Lupinus albus*.

**Figure 2 biomolecules-10-00292-f002:**
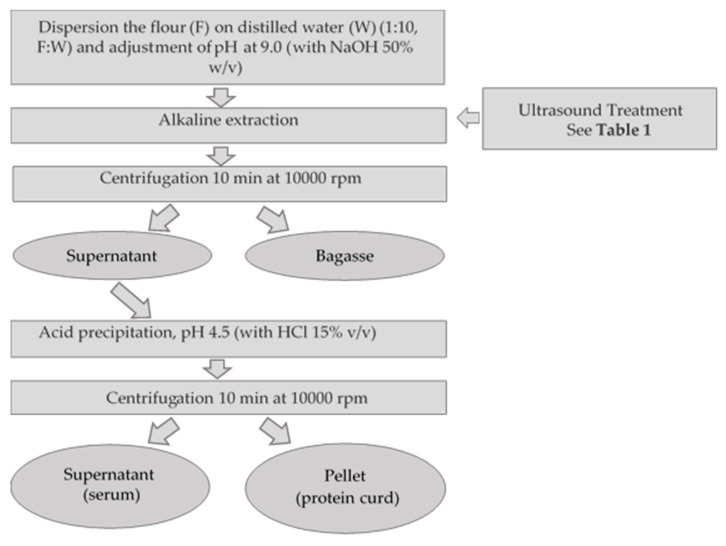
Procedure for alkaline extraction of flour lupine’s proteins assisted with ultrasound and acid precipitation.

**Figure 3 biomolecules-10-00292-f003:**
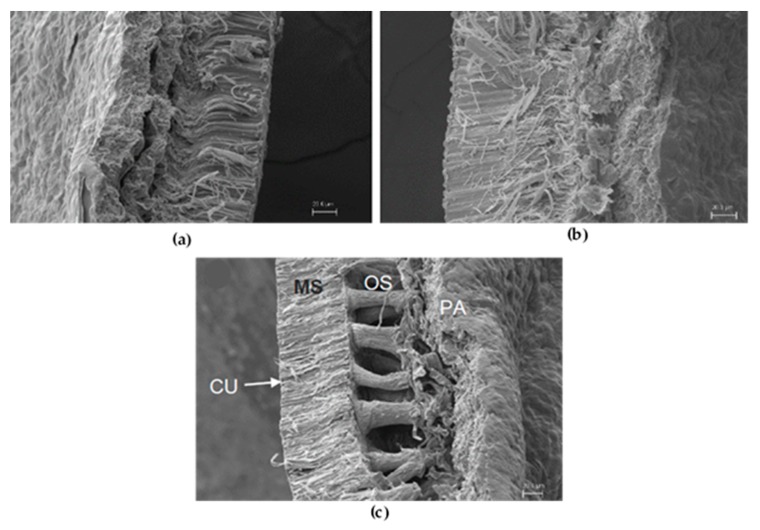
Scanning electron microscopy of *Lupinus* seeds coat of different varieties: (**a**) *L. albus*; (**b**) *L. angustifolius*; (**c**) *L. mutabilis*. CU-cuticle; MS-macrosclereids layer; OS-osteosclereid layer; PA-parenchymal cells.

**Figure 4 biomolecules-10-00292-f004:**
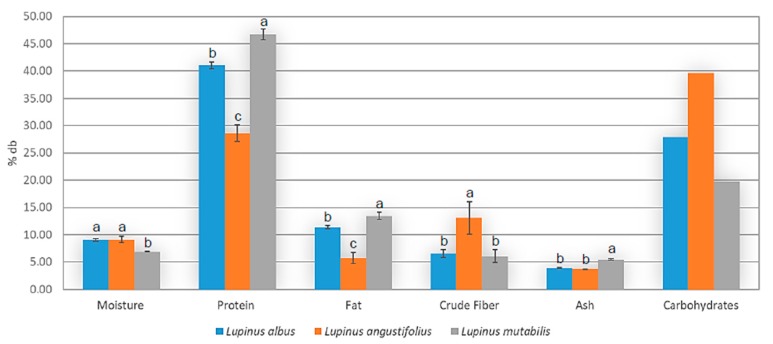
Proximate analysis (% db: dry basis percentage) of lupin flours of different varieties. Bars with different letters within proximate parameters are statistically different (*p* < 0.05).

**Figure 5 biomolecules-10-00292-f005:**
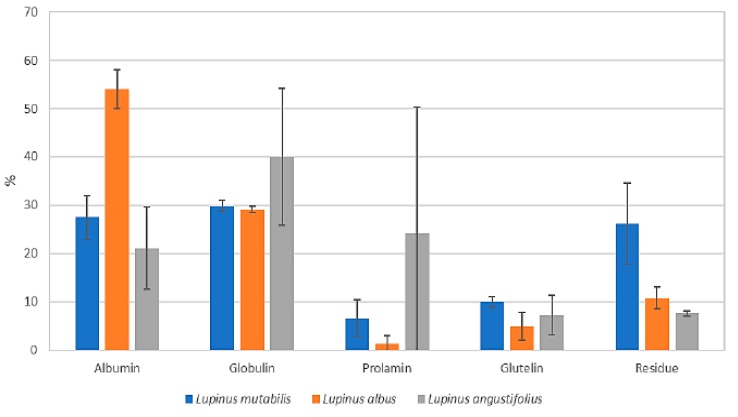
Osborne’s protein solubility profile (percentage, %) of lupin flour of different varieties.

**Figure 6 biomolecules-10-00292-f006:**
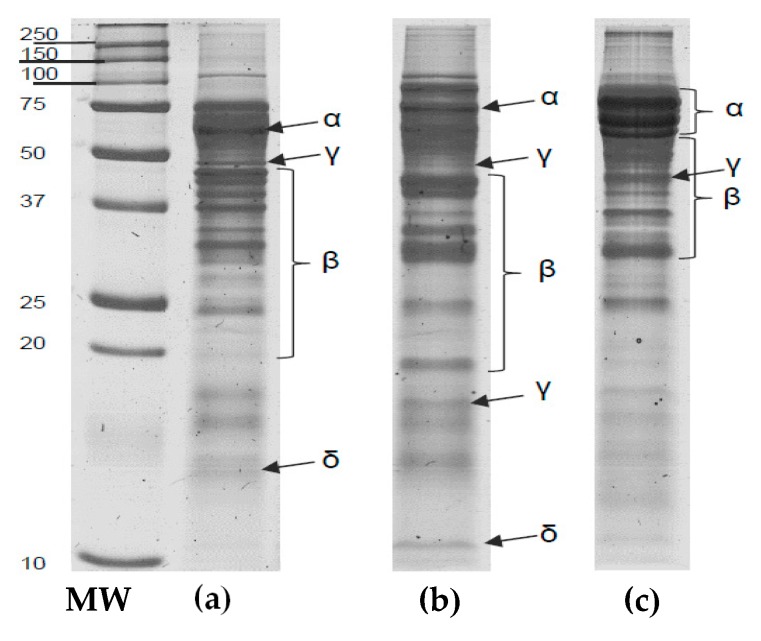
SDS-PAGE of three *Lupinus* varieties obtained in non-reducing conditions: (**a**) *L. albus*; (**b**) *L. angustifolius*; (**c**) *L. mutabilis*. MW- Molecular weight standard.

**Figure 7 biomolecules-10-00292-f007:**
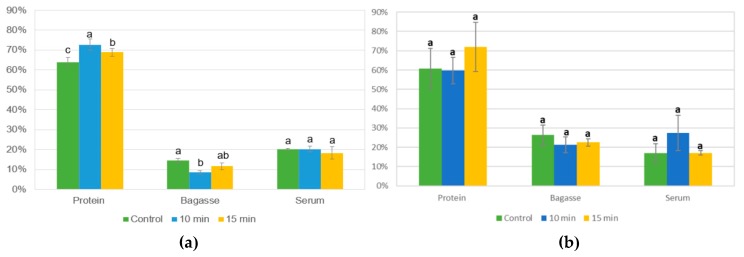
(**a**) *L. mutabilis* and (**b**) *L. angustifolius* protein extraction yield (%) for each fraction (protein curd, bagasse, and serum) obtained during the alkaline process assisted with US followed by isoelectric precipitation. Bars with different letters within evaluated fraction are statistically different (*p* < 0.05).

**Figure 8 biomolecules-10-00292-f008:**
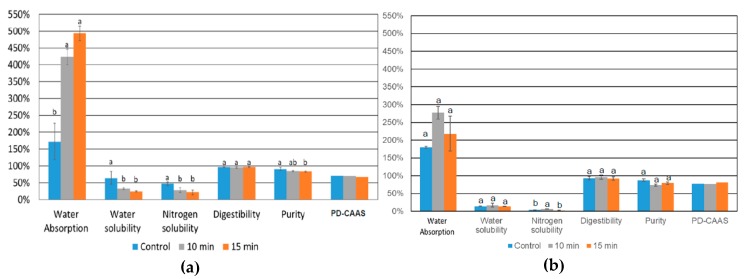
Functional properties of protein isolate from (**a**) *L. mutabilis* and (**b**) *L. angustifolius* obtained with alkaline process assisted with US. Bars with different letters within each functionality differ significantly (*p* < 0.05).

**Figure 9 biomolecules-10-00292-f009:**
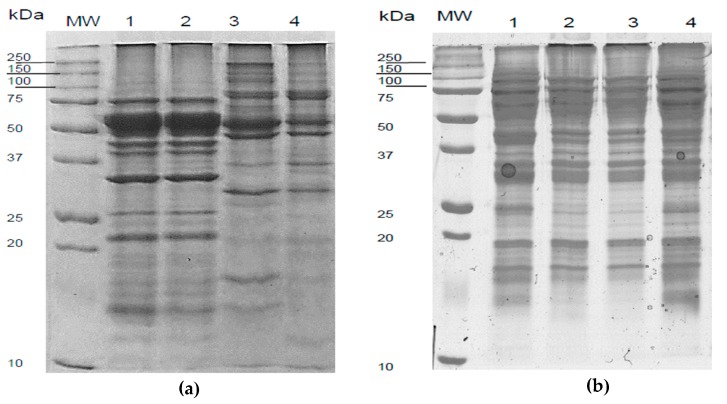
Electrophoretic profile for (**a**) *L. mutabilis* and (**b**) *L. angustifolius* protein extracts in non- reducing conditions. Line 1 is defatted flour; Line 2 is protein isolate; Line 3 protein isolate using 10 min US-assisted extraction and Line 4 protein isolate using a 15 min US-assisted extraction. MW-Molecular weight standard.

**Figure 10 biomolecules-10-00292-f010:**
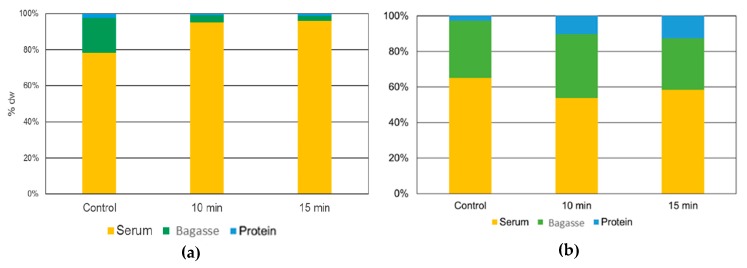
Quinolizidine alkaloids distribution in each fraction % of dry weight (serum, bagasse, and protein) obtained during US-assisted alkaline extraction and isoelectric precipitation from (**a**) *L. mutabilis* and (**b**) *L. angustifolius*.

**Table 1 biomolecules-10-00292-t001:** Experimental parameters used for alkaline protein extraction assisted with ultrasound using *Lupinus* flours as raw material.

Parameter	Unit	Value(s)
pH	-	9
Frequency	kHz	24
Temperature	°C	Monitored
Amplitude	μm	100
Sonotrode diameter	mm	22
Acoustic power density	W/cm^2^	85
Time	min	0, 10, 15

**Table 2 biomolecules-10-00292-t002:** Physical characteristics of *Lupinus* seeds for three different varieties: *albus*, *angustifolius,* and *mutabilis*.

Variety	Thousand Seed Weight (g)	Test Weight (kg/hL)	Foreign Material (%)	Damaged Kernels (%)	Foreign Seeds (%)
*Lupinus albus*	311.0 ± 13.64	60.23 ± 0.20	0.18	0.75	0.09
*Lupinus angustifolius*	140.1 ± 5.23	62.14 ± 0.42	0.95	0.36	0.35
*Lupinus mutabilis*	214.1 ± 0.90	59.01 ± 0.27	0.66	1.07	0.11

**Table 3 biomolecules-10-00292-t003:** Amino acid profile of lupin flours for three different varieties: *albus*, *angustifolius,* and *mutabilis*.

Amino Acid	*L. albus*	*L. angustifolius*	*L. mutabilis*
	(g/100 g of Crude Protein)		
Aspartic acid	10.52	10.35	10.36
Threonine	3.65	3.76	3.61
Serine	4.63	4.05	4.04
Glutamic acid	21.66	21.20	22.45
Proline	4.38	4.43	4.23
Glycine	4.07	4.65	4.25
Alanine	3.45	3.89	3.73
Cysteine	1.74	1.77	1.46
Valine	4.35	4.56	4.32
Methionine + Cys	2.50	2.58	2.20
Methionine	0.76	0.80	0.73
Isoleucine	4.71	4.56	4.82
Leucine	7.74	7.52	6.87
Tyrosine	4.32	3.59	4.20
Phenylalanine + Tyr	8.39	7.81	8.17
Phenylalanine	4.07	4.22	3.97
Lysine	5.02	5.62	5.93
Histidine	2.38	2.91	2.95
Arginine	10.86	10.56	10.67
Tryptophan	0.76	1.06	0.97
**Amino Acid Score ^1^**	**0.96**	**0.99**	**0.84**

^1^ The amino acid score is based on cysteine + methionine as limiting amino acid.

**Table 4 biomolecules-10-00292-t004:** Protein secondary structures (%) of *Lupinus* flours for three different varieties: *albus*, *angustifolius,* and *mutabilis*.

Variety	β-Sheet ^1^	3_10_ Helix	α-Helix	Unordered	β-Sheet	Aggregated Strands
*L. albus*	21.4	7.1	28.6	21.4	21.4	0.0
*L. angustifolius*	50.0	6.3	25.0	018.8	0.0	0.0
*L. mutabilis*	34.8	13.0	17.4	8.7	26.1	0.0

^1^ Antiparallel/aggregated strands.

**Table 5 biomolecules-10-00292-t005:** Amino acid profile of *L. mutabilis* and *L. angustifolius* protein isolates obtained with the different US-assisted alkaline extraction.

Amino Acid	Control (0 min)		10 min US Treatment		15 min US Treatment	
	(g/100 g of Crude Protein)					
	*L. mutabilis*	*L. angustifolius*	*L. mutabilis*	*L. angustifolius*	*L. mutabilis*	*L. angustifolius*
Aspartic acid	10.27	10.35	10.34	10.40	10.39	10.45
Threonine	3.24	3.12	3.43	3.59	3.43	3.62
Serine	5.05	4.69	4.22	4.05	4.20	3.98
Glutamic acid	24.08	24.15	21.78	20.88	22.03	20.59
Proline	4.17	4.41	4.37	4.57	4.37	4.52
Glycine	3.87	4.08	4.04	4.42	4.03	4.41
Alanine	3.25	3.23	3.53	3.77	3.52	3.78
Cysteine	1.25	1.60	1.14	1.41	1.11	1.42
Valine	4.09	4.08	4.59	4.79	4.53	4.88
Methionine+ Cys	1.82	2.16	2.82	2.17	1.74	2.21
Methionine	0.57	0.56	0.68	0.76	0.63	0.79
Isoleucine	5.01	4.69	5.25	4.91	5.22	4.98
Leucine	7.08	7.61	7.38	7.94	7.31	8.08
Tyrosine	3.81	3.63	4.26	3.93	4.24	3.92
Phenylalanine+Tyr	7.75	7.84	8.57	8.53	8.51	8.61
Phenylalanine	3.94	4.21	4.31	4.60	4.28	4.69
Lysine	5.34	4.61	5.59	5.31	5.59	5.31
Histidine	2.69	2.60	2.72	2.73	2.72	2.76
Arginine	11.28	11.29	11.14	10.47	11.19	10.37
Tryptophan	0.96	0.99	0.98	1.16	1.00	1.10
**Amino Acid Score ^1^**	**0.70**	**0.83**	**0.70**	**0.83**	**0.67**	**0.85**

^1^ The amino acid score is based on cysteine + methionine as limiting amino acid.

**Table 6 biomolecules-10-00292-t006:** Protein secondary structures of *L. mutabilis* protein isolates obtained during ultrasound-assisted alkaline extraction and isoelectric precipitation (%).

US Treatment	β-Sheet ^1^	3_10_ Helix	α-Helix	Unordered	β-Sheet	Aggregated Strands
Control (0 min)	40.00	20.0	20.0	0.0	10.0	10.0
10 min	50.0	0.0	12.5	12.5	12.5	12.5
15 min	57.1	0.0	14.3	14.3	0.0	14.3

^1^ Antiparallel/aggregated strands.

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
