# Peer review of "Effect of Ultrasound Application on Protein Yield and Fate of Alkaloids during Lupin Alkaline Extraction Process"

_biomolecules, 2020, doi:10.3390/biom10020292_

Round 1
Reviewer 1 Report
The present manuscript is focusing on the characterisation of the protein fate and yield during the alkaline extraction from three different Lupin species.
The authors should carefully check the English throughout the entire manuscript. Moreover, check that the Latin names are always written with italics.
Did the authors consider the milling conditions for the three materials' types (time, humidity, etc)? Depending on the conditions can be registered variations in the particle size.
What are the meanings of a, b, c in Figure 4?
It is advisable to maintain the colours in Figures 4 and 5.
In figure 5 is the error so big in case of prolamin (L. angustifolius)?
Please provide the standard deviation for the values presented in Table 3, as well as for the other results presented in the tables in the manuscript. Is it necessary to present the values obtained for the Methionine + Cys and Phenylalanine + Tyr?
Some abbreviations are not explained (NSI). These should be mentioned where they are first appearing.
Carefully check the way of writing 310 helix everywhere in the manuscript.
Author Response
Dear reviewer, thanks for your comments. Please see the attachment with the point-by-point reponses. The manuscript has improved a lot because of your suggestions, so thanks a lot for your time.
Regards,
Cristina

Reviewer 2 Report
Overall observation:- The topic in this Manuscript is one of the very interesting topics to the readers and to further investigations about Lupin flour protein.
Just some comments mainly about the manuscript regarding some clarification and suggestions to improve the manuscript. Mainely, about the presentation of the results and writing style.
1- Line 17: please provide an explanation of why only two species were used to study the behavior of the protein fraction....... Since the study included three species
2- The results section need to be improved
The author has used too much numerical value in results which is looking inappropriate. The author is advised that mention only the main key findings of tables and figures. Avoid using too many numerical values in it.
Line 405-406 : please move before Table 6 and combine with the sentence in 400-401
3- The discussion section is very lengthy
Using long sentences and unnecessary information can be very confusing to the readers.
for example 441-474. please shorten this part
Also Line 475-582: 4.4. Effects of the US treatment ....Please shorten with the focus on the most important findings
Line 430; It is evidence that....It is not clear what does the author want to establish
Line 620-623:- Please cite the reference of the statement QA concentrations ....
Author Response
Dear reviewer, thanks for your comments. Please see the attachment with the point-by-point reponses. The manuscript has improved a lot because of your suggestions. Thanks a lot for your time.
Regards,
Cristina

Reviewer 3 Report
In general, the article is well written. However, it requires some corrections and additions: 1) Normal concentration is not currently recommended by the International Union of Pure and Applied Chemistry. Therefore, authors should provide concentrations in mol / L. 2) Authors should attach Supplementary Materials with, among others GC-MS chromatograms. 3) Were the analytical methods used validated?Author Response
Dear reviewer, thanks for your comments. Please see the attachment with the point-by-point reponses. The manuscript has improved a lot because of your suggestions. Thanks a lot for your time.
Regards,
Cristina

Round 2
Reviewer 1 Report
The authors addressed all of the reviewer's recommendations.
The manuscript can be published in the present form.